# Assessment of the Operative Feasibility and Ventilation Distribution during Nonintubation Thoracoscopic Surgery Using Electrical Impedance Tomography

**DOI:** 10.3390/jpm12071066

**Published:** 2022-06-29

**Authors:** Kuan-Hsun Lin, Pei-Yi Chu, Zhanqi Zhao, Hung Chang, Po-Jen Yun, Tsai-Wang Huang

**Affiliations:** 1Division of Thoracic Surgery, Department of Surgery, Tri-Service General Hospital, National Defense Medical Center, Taipei 11490, Taiwan, China; dontchangela@gmail.com (K.-H.L.); chu.peiyi.88@gmail.com (P.-Y.C.); hung@mail.ndmctsgh.edu.tw (H.C.); jichinfish@gmail.com (P.-J.Y.); 2Institute of Technical Medicine, Furtwangen University, 78054 Villingen-Schwenningen, Germany; zhanqi.zhao@hs-furtwangen.de

**Keywords:** nonintubated thoracic surgery, ventilation, electrical impedance tomography

## Abstract

Background: To investigate the feasibility, ventilation distribution, and physiological effect of iatrogenic pneumothorax generated during nonintubated thoracoscopic surgery using electrical impedance tomography. Methods: Patients who underwent resections for pulmonary nodules between April 2016 and April 2019 were enrolled prospectively. Electrical impedance tomography was performed, and the measurements were recorded at five different timepoints. The patient characteristics, pathological characteristics, surgical procedures, operation times, and intraoperative parameters were recorded and analyzed. Results: Two hundred sixty-five perioperative electrical impedance tomography measurements during nonintubated thoracoscopic surgery were recorded in fifty-three patients. Fifty-one patients underwent wedge resections, and two patients underwent segmentectomies. The preoperative lateral decubitus position time point showed greater ventilation in the right lung than in the left lung. For left-sided surgery, the nonoperative lung had better ventilation (64.5% ± 14.1% for the right side vs. 35.5% ± 14.1% for the left side, *p* < 0.0001). For right-sided surgery, the nonoperative lung did not have better ventilation (52.4% ± 16.1% for the right side vs. 47.6% ± 16.1% for the left side, *p* = 0.44). The center of ventilation was significantly increased after surgery (*p* < 0.001). The global index of ventilation showed no difference after surgery. Conclusions: The nonintubated thoracoscopic surgical side had different ventilation distributions but reached ventilation equilibrium after the operation. Electrical impedance tomography is feasible and safe for monitoring ventilation without adverse effects.

## 1. Introduction

Since the advent of video-assisted thoracoscopic surgery (VATS), more thoracic operations have been performed, which include nonanatomical pulmonary resections and complicated sleeve procedures. Double-lumen endotracheal tube (DLET) and a bronchial blocker (BB) are the main tools used to achieve single-lung ventilation, which can provide an immobilized and steady surgical field intraoperatively. Using a DLET and a BB is the standard procedure for VATS, and the use of these tools has provided good surgical access. However, these tools often lead to complications and higher costs, and they require a bronchoscopy by an experienced operator [1,2]. The common complications are unsuccessful or difficult intubation, trauma of the airway, inappropriate device positioning intraoperatively, hypoxemia, bronchial injury, and vocal cord injury [2,3]. Given the associated disadvantages, there has been a resurgence in the use of nonintubated thoracic surgery (NITS) in recent years. The frequency of NITS continues to dramatically increase, and it has been proven in previous studies that it is feasible and safe in a variety of thoracic procedures, including pulmonary resections, the excision of pleural and mediastinal tumors [4] and complicated anatomical resections [5].

In NITS, collapse of the target lung is achieved by creating an open pneumothorax in the surgical hemithoracic region after making an intercostal incision. This provides good visualization and an excellent working field, but the operative lung also loses its ventilation function due to the pneumothorax, resulting in a ventilation-to-perfusion mismatch. In addition, air is exhaled from the spontaneously breathing nonoperative lung and goes into the operative lung during expiration. Part of this air is then re-inhaled back during inspiration, and this is termed “paradoxical respiration.” The atmospheric pressure in the open surgical hemithorax and pressure from the abdominal organs can also cause a downward shift of the mediastinum toward the surgical region. These phenomena can lead to decreased vital capacity, mismatched ventilation-to-perfusion, airway closure, atelectasis, hypercapnia, hypoxemia, and tachypnea, although the transient intermittent hypercapnia might be tolerated by the patients [6]. The long-term effects of these factors have not yet been comprehensively evaluated, especially from the perspectives of the oncological prognosis and physiological changes. All of the findings mentioned above are common events in NITS but need to be carefully considered, especially in regard to the patient’s respiratory pathophysiology.

Given that there are no real-time monitoring tools available for the exact regional distribution of ventilation when performing NITS, electrical impedance tomography (EIT) is considered. EIT is a noninvasive, real-time, and radiation-free bedside tool that enables the visualization of intraoperative regional ventilation distribution of ventilation. EIT has been widely utilized in previous studies for monitoring the regional ventilation, optimizing the positive end-expiratory pressure (PEEP) settings, and detecting overdistension and atelectasis [7], even for some critical situations, such as among patients with acute respiratory distress syndrome with extracorporeal membrane oxygenation [8]. EIT has also been used during surgical procedures, and it has shown promising results for adjusting the intraoperative ventilation strategy [9,10]. EIT has also been used to investigate the redistribution of the regional ventilation in patients who underwent open thoracic surgery and video-thoracoscopic procedures, and was associated with reduced ventilation and vital capacity on the surgical side after surgery [11]. NITS has now been widely adopted for thoracic surgery; however, the presence of transient hypercapnia, the real perioperative physiological changes, and the long-term effects of hypercapnia and hypoxia have not yet been studied in depth. Therefore, we conducted a pilot study using EIT throughout NITS procedures and evaluated whether EIT was feasible and could be used to monitor the distribution of ventilation between the lungs perioperatively.

## 2. Patients and Methods

This was a prospective study approved by the Institutional Review Board of Tri-Service General Hospital (TSGHIRB 2-106-05-105). Between April 2016 and April 2019, patients scheduled to undergo NITS (uniportal VATS [12]) for a sublobar pulmonary resection under spontaneous breathing with laryngeal mask airway (LMA) support were enrolled, and written informed consent was obtained from all the patients. The exclusion criteria were patients aged under 20 years, patients with an American Society of Anesthesiologists physical status class of III or greater, a body mass index (BMI) >30 kg/m^2^, an expected difficult airway management, chronic obstructive pulmonary disease or asthma leading to decreased lung function, large and central pulmonary lesions (>6 cm) for pulmonary resections, patients who were pregnant, patients who need emergency surgery, patients with a history of a previous ipsilateral chest surgery, chronic kidney disease higher than stage 3, liver disease with a Child–Pugh score >7, coagulopathy, and congenital or acquired oropharyngeal malformations and patients who have contraindications to the use of EIT (pacemaker, automatic implantable cardioverter defibrillator, and implantable pumps). Finally, 53 patients met the inclusion criteria and were included in the analysis.

The preoperative workups included computed tomography imaging of the chest, 3D simulation to reconstruct the pulmonary nodule (tumor size less than 2 cm and ground-glass opacity less than 0.5) with a 2-cm safe margin [13], general laboratory tests, cardiopulmonary function testing, and an anesthetic assessment. Propofol-based total intravenous anesthesia (TIVA) with sevoflurane anesthesia was applied as reported [14]. All patients fasted overnight before surgery, and there was no premedication before the induction of anesthesia. Intraoperative monitoring, such as measures of noninvasive arterial blood pressure, electrocardiography, pulse oximetry, end-tidal carbon dioxide pressure (EtCO_2_), and a bispectral index (BIS), was carried out for each patient. Anesthesia was induced with fentanyl and propofol in all patients and was then maintained with propofol or propofol/sevoflurane after the insertion of an LMA. Anesthesia was maintained using a target-controlled infusion (TCI) with a propofol infusion and spontaneous breathing with a 1.0 L/min flow rate (100% oxygen). The drug concentrations used for the TIVA were adjusted to keep the BIS value between 40 and 60 and the mean arterial pressure and heart rate within 20% of the baseline levels. The peripheral oxygen saturation (SpO_2_) was maintained at ≥90%. Incremental intravenous injections of fentanyl were administered if the patient developed moderate to severe coughing with limb movement (affecting the surgical procedure), and the patients’ respiratory rates were maintained at 12–20 breaths/min. Intercostal nerve blocks at the level of the incision were also applied to ensure complete analgesia. An ipsilateral vagal block was used to inhibit the cough reflex.

To record the EIT measurements, an EIT belt with 16 equidistantly integrated electrodes was applied around the patient’s thorax at the third to fourth intercostal space above the surgical incision, and a reference electrode was placed on the abdomen (PulmoVista 500, Dräger Medical AG, Lübeck, Germany; Figure 1). The optimal belt size was chosen according to each patient’s thoracic circumference. EIT data were recorded at five time points: supine position prior to surgery (event 1); lateral decubitus position prior to surgery (event 2); lateral decubitus position during surgery (standard VATS procedure with iatrogenic pneumothorax) (event 3); lateral decubitus position during surgery (standard VATS procedure without iatrogenic pneumothorax) (event 4); and supine position after surgery (event 5). (Figure 2) Each recording continued for at least 2 min. The measurement was discontinued when electrocautery was in use to avoid the machine being shut down by stray electric currents.

The EIT measurements were continuously recorded to track the changes in the pulmonary electrical bioimpedance. A low-pass filter was used to eliminate the cardiac-related impedance changes. The data were subsequently analyzed with vendor-supplied software (Dräger EIT Data Analysis Tool 6.1). The lung region of interest (ROI) was divided into four parts (ROI1, ROI2, ROI3, and ROI4) with equal height from ventral to dorsal. The EIT image matrix represents the impedance change relative to a baseline set at the tidal breathing during expiration at rest. The matrix data of the relative intensity changes were processed using Microsoft Excel. The 32 × 32 matrix variables were defined as follows: x = 1 (right) to 32 (left) and y = 1 (ventral) to 32 (dorsal). Within this matrix, the center of ventilation (COV) with sagittal (COVy) coordinates represented the center of gravity if each point of the matrix was assumed to represent a mass [15]. To allow the summation of the EIT images that were independent of the absolute impedance and breathing effort, each matrix of the tidal images was normalized to the same global impedance change. Thus, the sum of all the 32 × 32 intensity changes was the same for every EIT image. The averaged normalized matrices for each time point were used to visualize the pulmonary ventilation for a whole group by a colored contour line graph (Origin Pro 9.1 G, OriginLab Corporation, Northampton, MA, USA). Besides, the so-called global inhomogeneity (GI) index was calculated to assess the ventilation distribution [16].

### Statistical Analysis

Descriptive data are expressed as the mean ± standard deviation. Student’s t-test was used to investigate the continuous variables, and the χ^2^ test was used to compare the categorical variables between the groups. The COV was calculated to compare the preoperative event value (Ev. 1) with the value taken at the end of surgery (Ev. The Mann–Whitney nonparametric U test was used to compare the results for the intragroup comparisons with three or more time points. IBM SPSS Statistics (v. 18.0; IBM Corp., Armonk, NY, USA) was used for all analyses, and two-sided *p* values < 0.05 were considered statistically significant.

## 3. Results

Of the 53 patients enrolled, 26 underwent left-sided surgery, and 27 underwent right-sided surgery. Details of the basic patient characteristics, the characteristics of surgery and anesthesia, and the other parameters are presented in Table 1. A preoperative image-guided localization was conducted for all patients before surgery. The mean age of these patients was 58.30 years, and 34 (64.15%) patients had thoracic epidural anesthesia for pain control. The mean operation time was 63.4 min. Of these patients, 51 patients underwent wedge resection procedures for small pulmonary nodules, and two patients underwent segmentectomies. The mean BMI was 23.42 kg/m^2^. There were no conversions from NITS to intubation. All resected pulmonary specimens underwent intraoperative frozen sections. The final diagnosis used formalin-fixed and paraffin-embedded (FFPE) specimens. In total, 35 patients were diagnosed with primary lung cancers, and 2 patients were diagnosed with atypical adenomatous hyperplasia. In total, 16 patients were confirmed to have benign lesions. The mean hospital stay was 6.5 days. There were no surgical complications or mortalities. None of the patients complained about any discomfort caused by the EIT belt prior to anesthesia, and there were no EIT-related complications observed during the operations or before the patients were discharged. The characteristics of the patients who had left-sided NITS versus the characteristics of those who had right-sided NITS were not significantly different (Table 2).

The mean EIT measurement time was 11.8 min. The nonoperative lung was not always better ventilated. At Ev. 3, the ventilation in the right lung was significantly greater than that in the left lung (58.1% ± 16.2% for the right side; 41.9% ± 16.2% for the left side, *p* < 0.0001). Further analysis was conducted to separate the patients according to the side of surgery. In the patients with left-sided NITS, the ventilation was 64.5% ± 14.1% for the right-side ventilation versus 35.5% ± 14.1% for the left-side ventilation, *p* < 0.0001. The nonoperative lung had better ventilation. In contrast, in patients with right-sided NITS, the ventilation was 52.4% ± 16.1% for the right side vs. 47.6% ± 16.1% for the left; *p* = 0.44, showing that the nonoperative lung did not have better ventilation (Table 3). The COV and ventilation equilibrium before surgery (COV-Ev. 1 = 47.2% ± 5.2%) were significantly lower than after surgery (COV-Ev. 5 = 51.3% ± 4.0%), indicating that there was a ventilation redistribution toward the dorsal region at Ev. 5 compared with Ev. 1 (*p* < 0.0001). However, no significant changes were observed in the GI index at Ev. 1 or Ev. 5 (0.40 ± 0.10 vs. 0.38 ± 0.07; *p* = 0.2).

## 4. Discussion

The NITS is feasible and safe in a variety of thoracic procedures, including pulmonary resections and complicated anatomical resections [5]. The goal of NITS in thoracic surgery is to decrease each surgical procedure and anesthesia. However, there are contraindications for this method, including morbid obesity, a Mallampati rating of greater than II or different anatomical deviations and pathologies, extreme emphysema, pleural adhesions, severe moves of diaphragm and mediastinum, or non-compliant affected person. The overall conversion rate of NITS to general anesthesia ranges from 1% to 9% [17]. The common reasons for conversion include surgery-related events (e.g., coughing, severe emphysema, excessive adhesions, extreme movements of diaphragm, non-compliant patient, or bleeding) and anesthetic conditions (e.g., mediastinal movement, hypoxemia, or hypercapnia) [18]. However, there are no definitive criteria to select patients who can be admitted to undergo NITS procedures, although obesity is considered a contraindication. In NITS, collapse of the operated lung is achieved by creating an open pneumothorax in the surgical hemithorax, and this provides superb visualization and an excellent working field. As NITS requires extra experience, practice and vigilance in thoracic surgery, patients need to be cautiously chosen as candidates for the procedure. Limited surgical area of view can also appear, which patients include those with intense emphysema, excessive actions of the diaphragm, frequent coughing, or severe obesity. The operative lung loses its ventilation function during the NITS, which results in a ventilation-to-perfusion mismatch. The pathophysiology of iatrogenic pneumothorax and contralateral lung ventilation has not been studied before, and this is the first prospective study to evaluate the contralateral ventilation during an NITS procedure.

EIT is a noninvasive, real-time tool that enables the changes in the regional distribution of ventilation to be visualized intraoperatively. It has been widely utilized in previous studies for monitoring regional ventilation, optimizing the PEEP settings, and detecting overdistension and atelectasis [7]. Some studies have demonstrated the use of EIT-guided PEEP titration in patients undergoing laparoscopic abdominal surgery [10]. EIT was also used to investigate the redistribution of the regional ventilation in patients who underwent open thoracic surgeries and video-thoracoscopic procedures, and was associated with reduced ventilation and vital capacity on the surgical side after surgery [11]. In that study, 26 patients undergoing thoracic surgery were enrolled, and the EIT measurements were taken pre- and postoperatively. This demonstrated the feasibility and safety of EIT for lung surgery. In our study, EIT, which is a real-time consecutive monitoring tool, was used on patients undergoing NITS intraoperatively. In the NITS procedure, spontaneous breathing occurs in the nonoperative lung, and part of the air volume is inhaled and just fills the surgical lung. Both hypercapnia that can stimulate tachypnea and transient hypercapnia can generally be tolerated [6]. Hypoxia and hypercapnia in NITS might contribute to the oncologic outcomes. More studies are needed to clarify this issue. Here, based on our selection criteria, not all of the patients developed hypoxia and permissive hypercapnia that compromised the surgical procedure. In the future, we will try to determine the causes of hypercapnia based on the preoperative EIT ventilation distribution data.

In this study, we tried to clarify the changes in the physiological ventilation before, during, and after the operation. The nonoperative lung did not have better ventilation but might have been influenced by the effect of the heart space occupation. Before surgery, the COV-Ev. 1 was significantly lower than the later COV-Ev. 5, indicating that the ventilation distribution was toward the dorsal region. However, no significant changes were observed between COV-Ev.2 and COV-Ev.4 Thus, iatrogenic pneumothorax and spontaneous breathing during NITS did not compromise the patient’s ventilation after the closure of the surgical wound.

There were some limitations in this study. First, all enrolled patients had a normal BMI and few comorbidities. We do not know the feasibility of EIT during NITS for overweight patients or for COPD patients with compromised pulmonary functions. Second, the number of enrolled patients was small. Increasing the number of enrolled patients will be necessary to validate these preliminary results. Third, the airflow through the open wound might have influenced the EIT signal (intra- or extrapulmonary air). Fourth, in order to record the EIT measurements, the operative time had to be significantly prolonged. Fifth, the size of the resection of the lung was not recorded in the current study. Sixth, most patients received wedge resection. The application of EIT real-time monitoring the regional distribution of ventilation in NITS segmentectomy or lobectomy needs further study to clarify. Seventh, the collection of data might have been influenced by the variability of the shape, depth, and circumference of the different chests of the patients in the cohort. Further study is needed to clarify the role of extrapulmonary air for the signal interpretation.

## 5. Conclusions

EIT is a noninvasive, real-time, and radiation-free bedside tool that enables the visualization of intraoperative regional ventilation distribution of ventilation. EIT is feasible method to investigate the physiological changes after the formation of an iatrogenic pneumothorax in patients who underwent NITS (uniportal VATS).

## Figures and Tables

**Figure 1 jpm-12-01066-f001:**
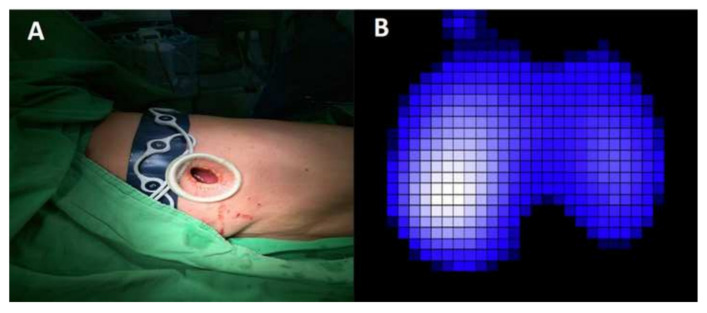
(**A**): The patient is placed in the right lateral position. The belt is applied to the third intercostal region. A uniportal VATS wound is made just below the EIT belt. (**B**): For left-sided NITS, the ventilation distribution at Ev.2 revealed better ventilation in the dependent lung (66.2% on the right vs. 33.8% on the left).

**Figure 2 jpm-12-01066-f002:**
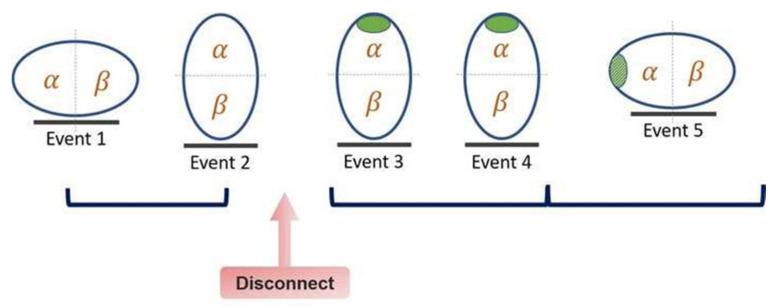
EIT measurement protocol at five different time points. Supine position prior to surgery (event 1); lateral decubitus position prior to surgery (event 2); lateral decubitus position during surgery (standard VATS procedure with iatrogenic pneumothorax) (event 3); lateral decubitus position during surgery (standard VATS procedure without iatrogenic pneumothorax) (event 4); and supine position after surgery (event 5).

**Table 1 jpm-12-01066-t001:** Characteristics of the patients undergoing NITS with EIT.

		*n* = 53
Age, year		58.30 ± 10.34
Gender, M/F (%)		19/34 (35.84)
BMI, kg/m^2^		23.42 ± 2.80
Thoracic epidural anesthesia, *n* (%)		34 (64.15)
Operation time, minutes		63.40 ± 24.3
Length of stay in hospital, days		6.45 ± 4.06
minSpO_2_, %		96.40 ± 2.15
maxEtCO_2_, %		53.68 ± 6.54
Surgical procedure, *n*		
	Wedge resection/segmentectomy	51/2
Complications related to EIT		
	Wound infection	0
	Interfere with the operation	0
	Skin burn/pressure sore	0
Pathology		
AAH, *n* (%)		2 (3.77)
Adenocarcinoma, n (%)		35 (66.04)
Benign, *n* (%)		16 (30.19)
Pulmonary function test		
	FEV1	79.4 ± 17.7
	DLCO	80.0 ± 12.0

**Table 2 jpm-12-01066-t002:** Patient Characteristics.

	Lt-Sided Sur (*n* = 26)	Rt-Sided Sur (*n* = 27)	*p* Value
Age, mean ± SD y	59.92 ± 8.72	56.67 ± 11.69	0.17
Female, *n* (%)	19 (73.1)	19 (70.3)	0.44
Height, mean ± SD cm	161.76 ± 8.89	161.37 ± 8.38	0.46
BMI, kg m^−2^	23.34 ± 2.57	23.49 ± 3.05	0.51
Pulmonary function test			
FEV1	80.7 ± 18.2	78.2 ± 17.5	0.612
DLCO	80.5 ± 10.1	79.6 ± 13.8	0.769
Thoracic epidural anesthesia, *n* (%)	16 (62)	18 (67)	0.39
Operation time, mean ± SD min	63.42 ± 24.32	73.85 ± 23.37	0.55
Length of stay in hospital, mean (IQR) d	7 (4.75–7)	6 (5–8)	0.29
minSpO_2_, median (IQR) %	97 (96–99)	96 (94–98)	0.63
maxEtCO_2_, median (IQR) %	53 (48–57)	55 (50–58)	0.87
Surgical procedure, n			
Wedge resection	26	25	
Segmentectomy	0	2	
Complications related to EIT			
Wound infection	0	0	
Interference with the operation	0	0	
Skin burn/pressure sore	0	0	
Belt size, n (S/M/L/XL) *	9/15/2/0	8/16/3/0	

Lt-sided Sur, left-sided surgery; Rt-sided Sur, right-sided surgery. * The belt sizes were selected according to the measured thoracic circumference.

**Table 3 jpm-12-01066-t003:** Ventilation distribution of the lungs bilaterally at Ev. 3 (the preoperative lateral decubitus position).

		Ventilation	
		Right better	Left better
Surgical side	Right	15 (58%)	11 (42%)
	Left	21 (87.5%)	3 (12.5%)

1 Case with equal ventilation in a Right-sided surgery, 2 Cases with missing data in a Left-sided surgery.

## Data Availability

The data used to support the findings of this study are available from the corresponding author upon request.

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
