# Peer review of "Assessment of the Operative Feasibility and Ventilation Distribution during Nonintubation Thoracoscopic Surgery Using Electrical Impedance Tomography"

_jpm, 2022, doi:10.3390/jpm12071066_

Round 1

Reviewer 1 Report

This is a prospective study of ventilation assessment in patients undergoing thoracic surgery by thoracoscopy using electrical impedance CT. It meets all ethical standards, but the authors must answer the following questions:
Is there a correlation of the degree of ventilation with the comorbidity of the patients?
Does the reliability of impedance CT to measure ventilation vary according to the type of resection?

Author Response

Reviewer #1:

This is a prospective study of ventilation assessment in patients undergoing thoracic surgery by thoracoscopy using electrical impedance CT. It meets all ethical standards, but the authors must answer the following questions:
Is there a correlation of the degree of ventilation with the comorbidity of the patients?
Does the reliability of impedance CT to measure ventilation vary according to the type of resection?

Reply1:

Thank you for the reviewer’s comment. We revised the statement in the manuscript.

Question 1: Is there a correlation of the degree of ventilation with the comorbidity of the patients?

Answer 1:

We agree there is a correlation of the degree of ventilation with the comorbidity of the patients.

In our previous experience, the NITS in the emphysematous lung with poor ventilation is extremely limited surgical area of view. In current study, the 53 patients without the comorbidity of chronic obstructive pulmonary disease such as emphysema and chronic bronchitis.

Question 2 Does the reliability of impedance CT to measure ventilation vary according to the type of resection?:

Answer 2:

In current study most patients receive wedge resection. The application of EIT real-time monitoring the regional distribution of ventilation in NITS segmentectomy or lobectomy is needed further study to clarify.

Reviewer 2 Report

Line 84

What method has been used for thoracoscopy? Was uniportal VATS performed, three ports VATS or single incision VATS. From the picture it seems that it is uniportal but in the text of figure 1A  the authors wrote single incision.

COMMENT: More informations are necessary.

Uniportal video-assisted thoracic surgery or single-incision video-assisted thoracic surgery for lung resection: clarifying definitions. Future Oncol2016; 12(23s), 6.

Line 87

The authors wrote: undergo NITS for a simple pulmonary resection

Comment: I agree for the wedge but certainly segmentectomy should not be defined a simple “resection”

Line 161

Why confidence interval have not been calculated? 

Line 178 There was no conversions from NITS to intubation. Thirty-five patients were diagnosed with primary lung cancers.

COMMENT: what did the authors do with these 35 patients with primary lung cancer. Was frozen section performed?

Line 216  In NITS, collapse of the operated lung 216 is achieved by creating an open pneumothorax in the surgical hemithorax, and this pro- 217 vides superb visualization and an excellent working field.

COMMENT: having performed many of these operations my data suggest that emphysematous lung have less chance to collapse. What is the authors experience? How many patients were emphysematous? Would you like to add few lines in the discussion? What are the physiologic changes in emphysematous lung?

I noted that there are many limitations written in the discussion. Do the authors think that this should become standard practice? What are the true advantages?

 Thank you for sending this paper to JPM surgery

Author Response

Reviewer #2:

Line 84

What method has been used for thoracoscopy? Was uniportal VATS performed, three ports VATS or single incision VATS. From the picture it seems that it is uniportal but in the text of figure 1A  the authors wrote single incision.

COMMENT: More informations are necessary.

Uniportal video-assisted thoracic surgery or single-incision video-assisted thoracic surgery for lung resection: clarifying definitions. Future Oncol, 2016; 12(23s), 6.

Reply1:

Thank you for the reviewer’s comment.

All surgical interventions in the current study are uniportal VATS. The manuscript was revised and corrected

Line 87

The authors wrote: undergo NITS for a simple pulmonary resection

Comment: I agree for the wedge but certainly segmentectomy should not be defined a simple “resection”

Reply2:

Thank you for the reviewer’s comment.

Segmentectomy in not a simple “resection”. The manuscript was revised and corrected as sublobar pulmonary resection.

Line 161

Why confidence interval have not been calculated? 

Reply3:

Thank you for the reviewer’s comment.

In this study, we use descriptive data are expressed as the mean ± standard deviation. Student’s t test was used to investigate the continuous variables, and the c2 test was used to compare the cate-gorical variables between the groups. IBM SPSS Statistics (v. 18.0; IBM Corp., Armonk, NY, USA) was used for all analyses, and two-sided P values <.05 were considered statistically significant.

Line 178 There was no conversions from NITS to intubation. Thirty-five patients were diagnosed with primary lung cancers.

COMMENT: what did the authors do with these 35 patients with primary lung cancer. Was frozen section performed?

Reply4:

Thank you for the reviewer’s comment.

All resected pulmonary specimens underwent intraoperative frozen sections. The final diagnosis using formalin-fixed and paraffin-embedded (FFPE) specimens. The manuscript was revised and corrected as sublobar pulmonary resection.

Line 216  In NITS, collapse of the operated lung 216 is achieved by creating an open pneumothorax in the surgical hemithorax, and this pro- 217 vides superb visualization and an excellent working field.

COMMENT: having performed many of these operations my data suggest that emphysematous lung have less chance to collapse. What is the authors experience? How many patients were emphysematous? Would you like to add few lines in the discussion? What are the physiologic changes in emphysematous lung?

Reply5:

Thank you for the reviewer’s comment.

In our previous experience, the NITS in the emphysematous lung is extremely limited surgical area of view. In current study, the 53 patients without the comorbidity of chronic obstructive pulmonary disease such as emphysema and chronic bronchitis.

We revised the manuscript LINE 217 and LINE 230 as following

LINE 217

The NITS in thoracic surgery goal to decrease each surgical procedure and anesthesia. Still, there are contraindications for this method inclusive of morbid obesity, a Mallampati rating of greater than II or different anatomical deviations and pathologies, extreme emphysema, pleural adhesions, severe moves of diaphragm and mediastinum, or non-compliant affected person.

LINE 230

As NITS require extra experience, practice and vigilance in thoracic surgery, patients need to be cautiously decided on as candidates for the procedure. Limited surgical area of view can also additionally appear, which patients include intense emphysema, excessive actions of the diaphragm, frequent coughing , or severe obesity.

I noted that there are many limitations written in the discussion. Do the authors think that this should become standard practice? What are the true advantages?

Reply6:

Thank you for the reviewer’s comment.

In our previous experience, the NITS in the emphysematous lung is extremely limited surgical area of view. In current study, the 53 patients without the comorbidity of chronic obstructive pulmonary disease such as emphysema and chronic bronchitis.

Round 2

Reviewer 2 Report

Dear Authors 

thank you for your answers. There are 3 main points that should be changed and added. The suggested reference should be included in the list and

1) the most important is why the authors after the positive frozen section for cancer did not performed segmentectomy or lobectomy. A clear explanation is necessary.

2) the "true" clinical message which can be extrapolated from this study should be given to the readers.

3) the reference for the uniportal VATS vs single incision VATS should have been added.

Thank you for sending this paper to JPM 

Author Response

Question 1: the most important is why the authors after the positive frozen section for cancer did not performed segmentectomy or lobectomy. A clear explanation is necessary.

Reply 1: Thank you for the reviewer’s comment. I have revised the manuscript. In the study, the tumor size was measured in the largest cross-section by averaging the length and width. Ground-glass opacity (GGO) was defined as an area with a homogeneous slight increase in density that did not obstruct the underlying vascular signal. All resected tumor was less than 2cm and GGO ratio less than 0.5. We performed preoperative simulation to satisfy the requirement for a sufficiently safe spherical margin greater than the nodule size or 20 mm in size. We add a new reference as following.

Lin, K.H., et al., Benefit of Three-dimensional Image Simulation in Surgical Resection of Early-stage Lung Cancer. Ann Thorac Surg, 2021.

Question 2: the "true" clinical message which can be extrapolated from this study should be given to the readers.

Reply 2: Thank you for the reviewer’s comment. I have revised the conclusion. EIT is a noninvasive, real-time, and radiation-free bedside tool that enables the visualization of intraoperative regional ventilation distribution of ventilation.  EIT is feasible method to investigate the physiological changes after the formation of an iatrogenic pneumothorax in patients who underwent NITS.

Question 3: the reference for the uniportal VATS vs single incision VATS should have been added

Reply 3:Thank you for the reviewer’s comment. I have added the reference to the manuscript.LINE 90.

Thanks again for all the effort you and your team done for me.

I have tried my best to revise the manuscript.

Sincerely,